# Recurrent Episodes of Acute Myocardial Infarction Secondary to Paradoxical Coronary Artery Embolism

Mita Singh [1,*], Ana Teresa Gomes [2], Paul Hill [2] and Ansuman Saha [2]

1    Lewisham and Greenwich NHS Trust, London SE18 4QH, UK
2    Surrey and Sussex NHS Healthcare Trust, Redhill RH1 5RH, UK; anateresa.gomes@nhs.net (A.T.G.);
     paul.hill3@nhs.net (P.H.); ansuman.saha@nhs.net (A.S.)
*    Correspondence: mita.singh@nhs.net

**Abstract:** Coronary artery embolism is a rare cause of acute myocardial infarction, attributed to approximately 10% of all paradoxical embolisms. It is a condition that should be considered in patients who present with chest pain and have a low overall risk of coronary heart disease. A major risk of coronary artery embolism is the existence of a patent foramen ovale (PFO), which can be shown on bubble transthoracic echocardiography. Here we describe a case report of a 68-year-old Caucasian lady who presented with recurrent episodes of myocardial infarction secondary to a paradoxical coronary artery embolism which was likely due to a PFO. We emphasize the need for more research on the role of PFO percutaneous device closure compared to just medical therapy in those with recurrent episodes of acute myocardial infarction secondary to paradoxical coronary artery embolism. This, in turn, should provide clearer guidance in managing such patients with high risk of mortality.

**Keywords:** patent foramen ovale; coronary artery embolism; stroke





## 1. Introduction

A paradoxical embolism is when an embolus originates in the systemic venous circulation and then enters the systemic arterial circulation through a cardiac wall malformation such as a patent foramen ovale (PFO) [1]. A PFO is effectively a left to right shunt between the left and right atria. In circumstances where the right atrial pressure is increased (i.e., Valsalva manoeuvres such as coughing or defecating), there is a brief reversal of the shunt, leading to the possible transmission of the thrombi from the right atrium to the left atrium via the PFO [2]. Furthermore, paradoxical emboli are suspected in these scenarios when patients have thromboembolic risks factors (such as atrial fibrillation) but no obvious source [3]. Of all of the cases of paradoxical embolism, approximately 10% are attributed to embolism of the coronary artery—an under-diagnosed and atypical cause of acute myocardial infarction (MI) [4]. As with our case, it should be considered in patients who present with chest pain and possess a low overall risk of coronary heart disease [1].

## 2. Case Presentation

A 68-year-old slim Caucasian lady of normal body mass index (height: 170 cm, weight: 55 kg), presented to East Surrey Hospital (November 2019), admitted with chest pain and sub-sternal chest pressure. Her ambulatory electrocardiogram (ECG) outlined ST segment elevation in leads V3-V4 (Figure 1A). She described the pain to be sharp in nature and central, which started gradually at work. She reported getting similar pains when gardening, and that the pain was accompanied with dizziness and palpitations. Furthermore, she has a background of diverticulitis, no history of venous thrombosis (although family history of stroke), does not smoke, or suffer from diabetes or hypertension and eats a balanced diet.

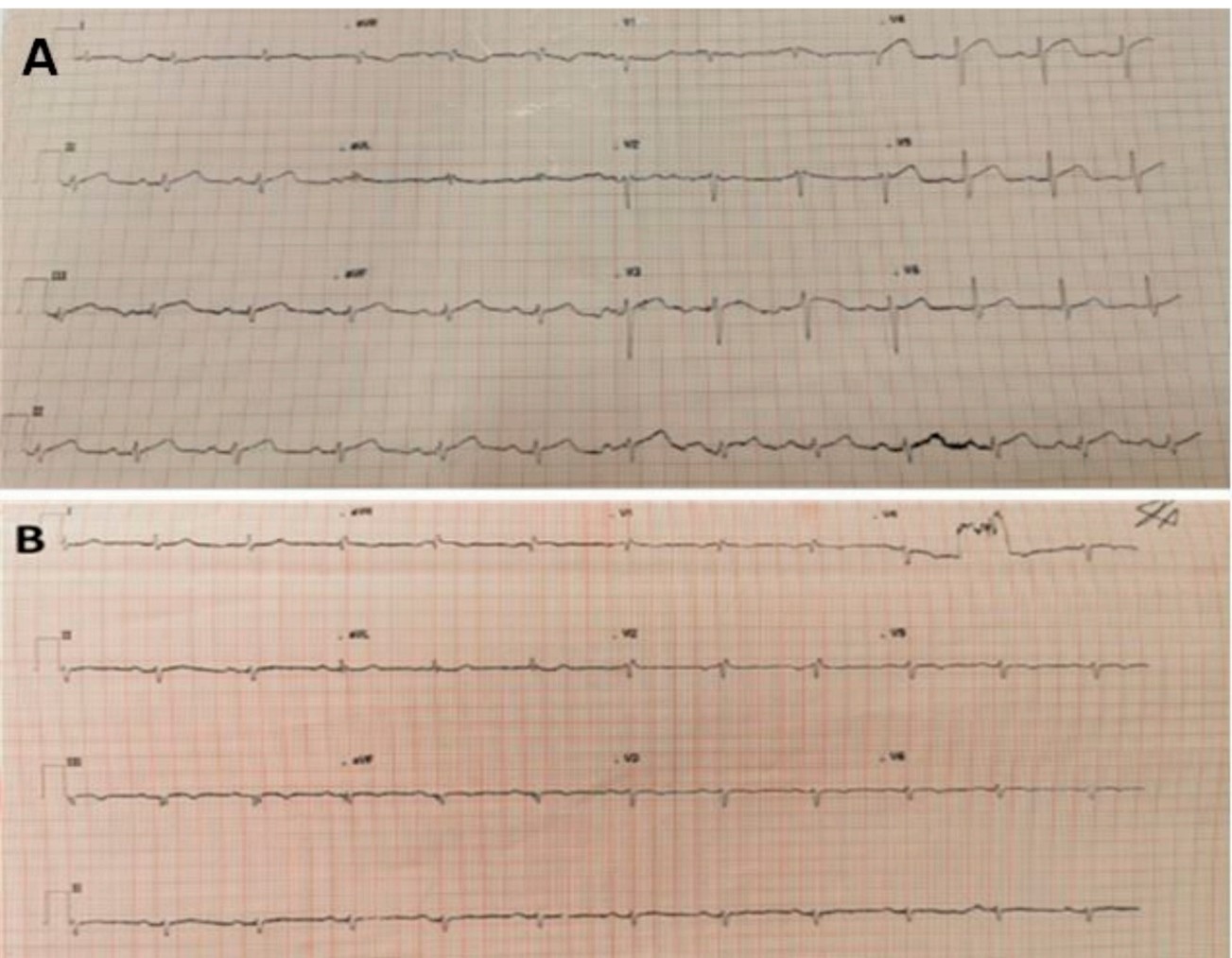

**Figure 1.** (**A**) 12-lead ECG taken on first admission. Sinus rhythm with ST segment elevation in chest leads V3-4 and limb lead II. Heart Rate 74, PR 210 ms, QRS 109 ms, QT/QTc 408/436 ms. (**B**) 12-lead ECG taken on second admission to A and E, which outlined sinus rhythm with right bundle branch block and T wave inversion from V3 to V6; VR 69, PR interval 185 ms, QRS 118 ms, QT/QTcinterval—384/403 ms.

On examination, her pulse was 90 beats per minute (and irregular), jugular venous pressure was unremarkable and heart sounds were normal except for ectopic beats. No murmurs or pericardial rub was noted, auscultation findings were normal, and she did not exhibit any calf tenderness.

## 3. Investigations, Results and Treatment

Her blood samples were unremarkable (a previous measure of low-density lipoprotein was also normal), with serial troponin levels being 25 to 28 to 587 (Table 1). Emergency bedside echocardiography found normal biventricular function (left ventricular ejection fraction > 60%) with no regional wall motion abnormality (RWMA) or valvular disease detected; chest X-ray also appeared normal (Figure 2A). Subsequently, she underwent an urgent coronary angiogram, which outlined tortuous unobstructed coronaries (Figure 3), and this result was also reviewed by accompanying cardiologists. It was therefore decided to admit her for a formal transthoracic echocardiogram (post-angiogram), inpatient cardiac magnetic resonance imaging (MRI) and 24-h ECG tape, and to check D-dimers levels whilst also stopping acute coronary syndrome protocol treatment.

**Table 1.** Hematology results taken on both admissions.

| Value | 04/11/19 | 05/11/19 | 06/11/19 | 07/11/19 | 11/11/19 | 12/11/19 | 13/11/19 | 14/11/19 | 17/01/20 | 18/01/20 | 19/01/20 | 20/01/20 | 26/01/20 | 17/02/20 |
|---|---|---|---|---|---|---|---|---|---|---|---|---|---|---|
| Hb | 152 H | 138 | 156 H | 162 H | 145 | 130 | - | - | 147 | - | - | 135 | 138 | 153 H |
| WBCC | 10.4 | 8.5 | 9.3 | 7.6 | 6.8 | 7.1 | - | - | 9.9 | - | - | 9.3 | 6.8 | 9.1 |
| PLT | 228 | 245 | 265 | 258 | 278 | 250 | - | - | 254 | - | - | 238 | 258 | 312 |
| INR | 1.0 | 1.1 | 1.0 | - | - | - | - | - | 1.1 | - | - | - | 1.4 | 1.1 |
| DD | - | 0.27 | - | - | - | - | - | - | - | - | - | - | - | 0.27 |
| Na | 140 | - | 143 | 142 | 140 | - | - | - | 138 | - | - | 138 | 139 | 142 |
| K | 4.3 | - | 4.0 | 4.1 | 4.1 | - | - | - | 4.1 | - | - | 4.4 | 4.4 | 5.3 H |
| Urea | 4.1 | - | 2.4 | 3.0 | 2.9 | - | - | - | 3.3 | - | - | 2.9 | 1.9 L | 4.0 |
| Cr | 58 | - | 69 | 65 | 56 | - | - | - | 47 | - | - | 45 | 54 | 59 |
| eGFR | 90 | - | 73 | 79 | >90 | - | - | - | >90 | - | - | >90 | >90 | 88 |
| CRP | - | - | <1 | <1 | <1 | - | - | - | <1 | - | - | <1 | <1 | - |
| Trop | 25–28 H | - | 587 H | 461 H | 130 H | 86 H | 48 H | 36 H | 173 H | 1017 H | 532 H | 375 H | - | 7 |

Abbreviations: Hb—haemoglobin, WBCC—white blood cell count, PLT—platelet, INR—internationalized ratio, Na⁺—sodium, K⁺—potassium, Cr—creatinine, eGFR—estimated glomerular filtration rate, CRP—C-reactive protein; H—higher than normal range, L—lower than normal range. The red borderline delineates the blood results taken from the patients' first and second admission.

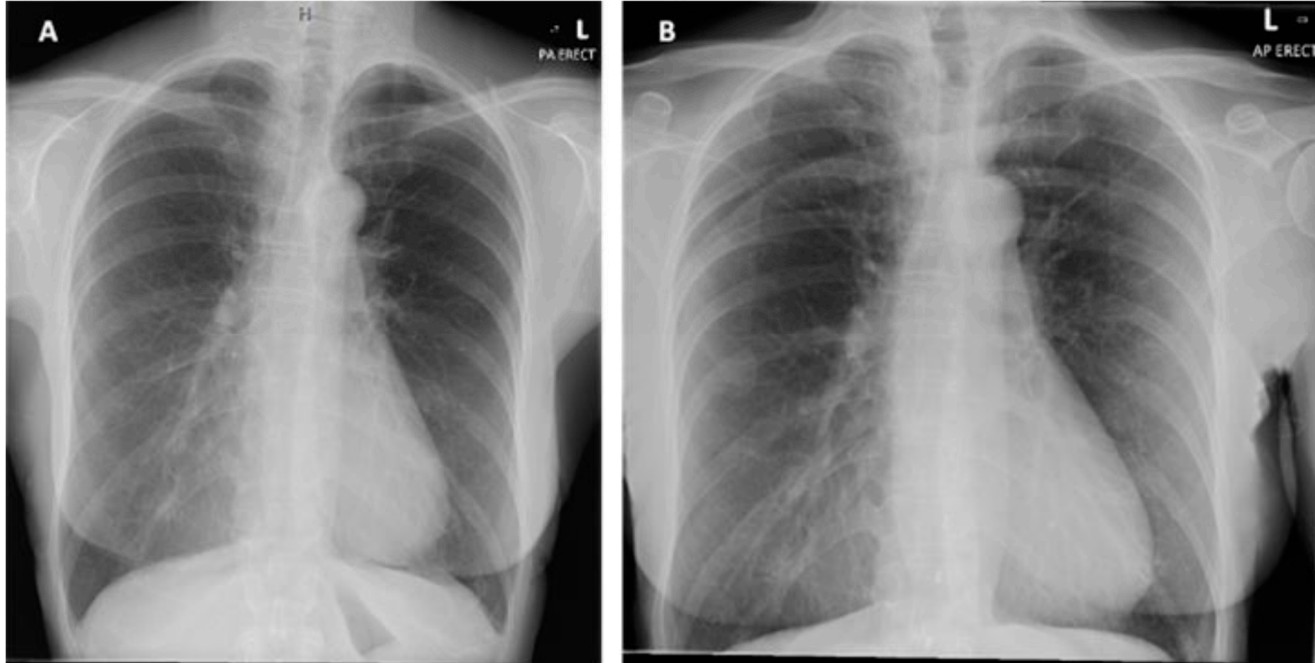

**Figure 2.** (**A**) Posterior-anterior erect X-ray taken on initial admission in November 2019. Normal cardiothoracic ratio, normal heart size and lungs are clear with no focal abnormality. (**B**) PA erect X-ray taken on second admission in January 2020. No new features compared to previous X-ray.

The formal echocardiogram confirmed similar findings to that carried out by the bedside, with further results found. Hypokinesia of the mid to apical inferior segments was found along with a bright myocardium. Grade 1 diastolic dysfunction was also noted. Moreover, a bright pericardium was noted with no evidence of pericardial effusion. Also, there was no obvious flow across the interatrial septal wall. Despite that, her transthoracic echocardiographic images could not exclude the possibility of a PFO. Whilst her D-dimer levels came back as normal, her 24 h ECG tape confirmed five episodes of atrial fibrillation (min heart rate 90, max heart rate 146), with each episode separated by short intervals of sinus rhythm (atrial fibrillation lasted for 4 h and 43 min in total). Furthermore, her cardiac MRI confirmed preserved global left ventricle systolic function with hypokinesia at the apex (Figure 4A–C). A small apical infarction was noted with one non-viable segment (out of a total of seventeen for the entire left ventricle). As such, her admission of acute

apical myocardial infarction (MI) was thought to be caused by an embolic phenomenon, precipitated by either paroxysmal atrial fibrillation (pAF) or a possible PFO. In order to delineate the cause, a bubble echocardiogram was carried out (Figure 5), something which is considered as a useful investigative tool for diagnosing cardiac wall malformations [5]. Subsequently, a reasonably sized PFO was found, wherein moderate amounts of bubbles were seen crossing the intra-atrial septum during normal breathing and during Valsalva maneuver (Figure 5). She was therefore discharged on aspirin 75 mg once daily, bisoprolol 1.25 mg twice daily, edoxaban 30 mg once daily (later doubled to 60 mg once daily) lansoprazole 30 mg once daily, and ramipril 1.25 mg once nightly. She was also due for outpatient work-up for PFO closure at a tertiary Centre.

Despite that, two months later, she reattended the Accident and Emergency Unit with a second episode of chest pain. An ambulatory ECG outlined poor R wave progression and T wave inversion in the lateral leads (Figure 1B). Her troponin was again elevated from 173 as initial and on repeat (Table 1), 1017 to 532 to 375. Her vital signs were however unremarkable (heart sounds 1 and 2, heart rate 77, chest clear, no calf tenderness). Her chest X-ray was also unchanged (Figure 2B). A 24 h ECG tape found her to be in sinus rhythm throughout (mean 71 bpm). However, compared to her previous cardiac MRI, a repeated scan exemplified a new inferolateral full thickness MI (Figure 4D) with myocardial edema and microvascular obstruction of two affected segments (with clear RWMA). This is in addition to her previous mature full thickness apical septal MI (2 segments), and despite these developments, her systolic function remained preserved. A complete thrombophilia screen was not carried out.

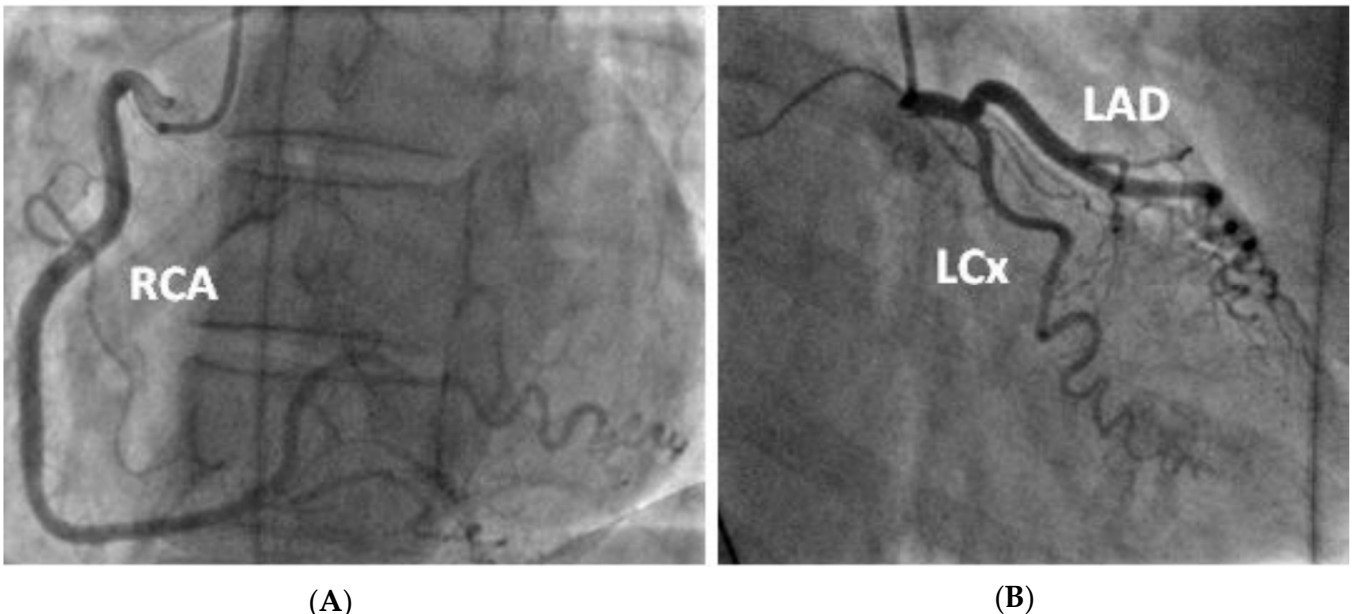

(**A**)                                                                                                      (**B**)

**Figure 3.** Coronary angiogram demonstrating tortuous unobstructed coronaries. —(**A**) LAO straight view of the right coronary artery after injection of contrast (**B**) RAO caudal view of the left coronary artery (including the left main stem, left anterior descending and left circumflex artery).

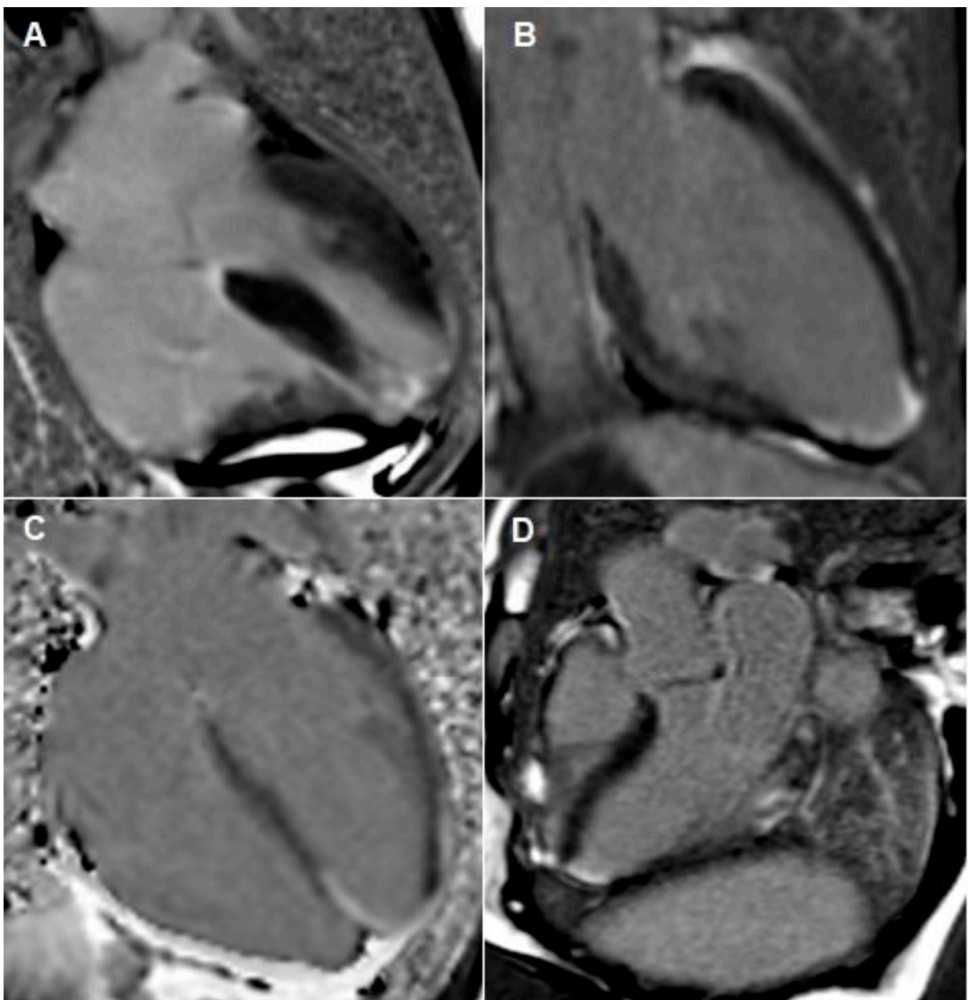

**Figure 4.** Cardiac MRI images taken during both admissions. (**A–C**) outlines apical infarction. These images outline focal transmural late gadolinium enhancement in apical septum with sub-endocardial myocardial late gadolinium enhancement extending into the distal apical anterior, apical inferior and true apical segments. Image (**D**) outlines inferior-lateral infarction, which occurred during her second admission.

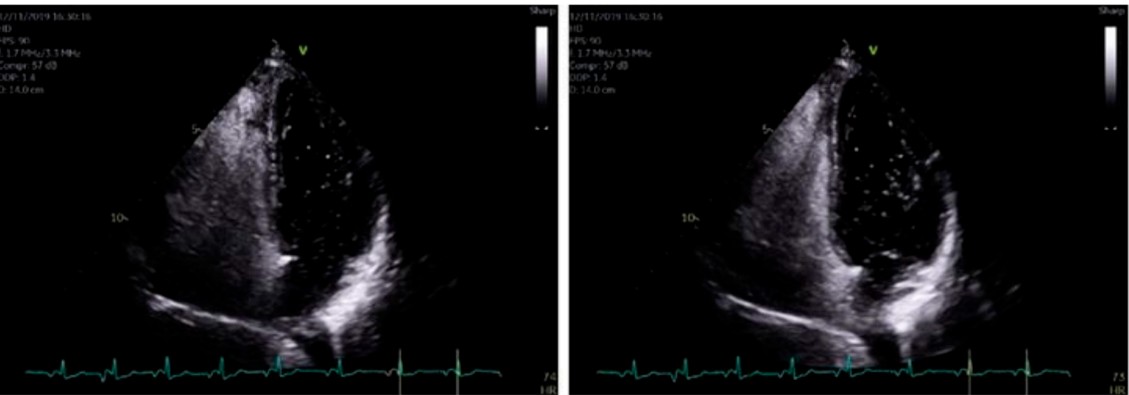

**Figure 5.** When using bubble transthoracic echocardiogram, bubbles will be visualized entering the right heart using the apical four-chamber or subcostal views. Once there is complete opacification of the right atrium (RA), cardiac cycles are counted [5]. In patient without any type of shunt present, no bubbles should appear in the left heart. If, bubbles appear in the left atrium (LA) after three cardiac cycles, an intracardiac shunt is likely present.

## 4. Outcome and Follow-Up

Upon further review, she is now awaiting urgent PFO closure at the tertiary Centre on the account of another MI caused by a suspected paradoxical coronary artery embolism.

## 5. Discussion

In 1877, Cohnheim was the first to describe a paradoxical embolism [6]. It is known to cause occlusion of the cerebral, peripheral arterial, and, in much more unique circumstances, of a coronary artery [7]. This explains the wide variety in presentations described by other case reports [8,9]. A major criterion in diagnosing coronary artery embolism includes the presence of an abnormal pathway between the venous and arterial circulation such as a PFO [10]. Furthermore, our patient had paroxysmal atrial fibrillation which may have acted as a source of emboli. Whilst she exhibited normal unobstructed coronary arteries, the presence of a PFO does run the risk of recurring episodes of acute MI (as we found with our patient).

Furthermore, when comparing our case with other published case reports, such as Hakim et al., 2014, Jamiel et al., 2012 and Boumaaz et al., 2020, they further concluded that paradoxical emboli should be suspected in patients with low risk profile for coronary heart disease [1,11,12]. However, more interestingly, different treatment methods were used in these cases to mitigate against the risk of recurrent MIs [1,11,12]. Some used anticoagulation (where percutaneous device closure was not indicated), whilst others opted for percutaneous device closure of the PFO.

Whilst PFO percutaneous device closure has been hypothesized as the better therapeutic option in patients at risk of recurrent MIs, there are very limited data to support this [13]. A critical review published in 2019 compared several randomized controlled trials to discuss the role of percutaneous device closure vs. anticoagulation in cases of paradoxical embolism. The review highlighted that the vast majority of these trials used stroke or transient ischaemic attacks as their endpoint as opposed to acute MIs. Additionally, they supported the role of PFO percutaneous device closure in stroke (where possible) [13]. As such, there is limited evidence to support the role of surgery to prevent recurrent MIs in patient with PFOs. In contrast, however, the review did suggest that percutaneous device closure would deem sensible in patients with ST segment elevation MI and normal coronary arteries [13].

Furthermore, guidance set by the National Institute of Clinical Excellence (NICE), reports that "the optimal treatment for patent foramen ovale in patients who have had a thromboembolic event remains undefined" [14]. Guidelines on the management of recurrent episodes of acute MI, also remain scarce [14]. This is in contrast to recurrent episodes of stroke, where other case reports of PFO have referred to national guidance, suggesting that percutaneous device closure should be considered [8].

In conclusion, there is limited evidence available to discuss the role of PFO closure in those at risk of recurrent acute MIs secondary to paradoxical coronary artery embolism. As such, this case highlights the necessity for further research and, thereby, clearer guidance on managing such patients with high risk of mortality.

**Author Contributions:** M.S.—involved with writing the report and liaising with the other authors to discuss the case. A.T.G.—involved with providing transthoracic echocardiographic images with captions describing the figures. P.H.—involved with proofreading and providing additional input in composing the case report. A.S.—providing interpretations to the cardiac MRI images and proofreading the case report. All authors have read and agreed to the published version of the manuscript.

**Funding:** This research received no external funding.

**Institutional Review Board Statement:** Not applicable.

**Informed Consent Statement:** Written informed consent has been obtained from the patient to publish this paper.

**Conflicts of Interest:** The authors declare no conflict of interest.

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
