# Peer review of "Recurrent Episodes of Acute Myocardial Infarction Secondary to Paradoxical Coronary Artery Embolism"

_cardiogenetics, doi:10.3390/cardiogenetics12030023_

Round 1

Reviewer 1 Report

They have done a great job, the publication will give insight in patients with PFO presenting with chest pain despite having a normal CAG, we should investigate more. And give them a proper Management

Author Response

Many thanks for your comments, we had added some further comparisons to other case reports varying between treatment options for mitigating against the risk of recurrent MIs in these patients. This case reports further highlights this disparity in treatment and the lack of further research in this field

Reviewer 2 Report

Introduction

The introduction should have more information about the pathology being addressed. Two references are too few to describe this topic.

In the discussion, the results obtained should be compared more extensively with the other published clinical cases as well as reviews on this subject.

The description of the case needs to go deeper into the case presented. What specific BMI did he have? Did the patient have a family history? These questions are necessary to present the topic correctly, as well as to include that information in some supplementary material.

Author Response

Many thanks for your comments. 

The introduction has now been bolstered with further detail on pathophysiology. 

The discussion has been further consolidated with comparisons made to other case reports, highlighting the "type" of patient likely to suspect for paradoxical emboli. We have also highlighted the disparity in treatment methods in managing recurrent MIs between these cases. We have also referenced a large review comparing several RCTs in identifying treatment methods in recurrent embolic events in such patients. Most often the end-points are stroke/TIA.

We have also outlined the family history and the patient had a normal BMI. 

Round 2

Reviewer 2 Report

The authors have done a better job that provides their manuscript with greater scientific rigor

Author Response

Many thanks for the feedback.